# Remote monitoring, personnel extinguishment or helicopter extinguishment? How to control forest fires more effectively

**Yuntao Bai[1], Lan Wang[2]\*, Xiaolong Yuan[1]**

1 Business School, Shandong Management University, Jinan, China, 2 Center of Emergency Management, Chongqing Academy of Governance, Chongqing, China

\* wanglan-8722@hotmail.com

**Data Availability Statement:** All relevant data are within the paper.

**Funding:** This research was funded by the Doctoral Research Foundation of Shandong Management University (sdmud2023001) and Social Science

## Abstract

Forest fires have a great impact on the ecological environment. The government needs to take measures to control forest fires. Forest fires are easily affected by wind speed and other weather conditions, and the difficulty of extinguishing forest fires is easily affected by terrain complexity. Effective management methods can protect the important ecological functions of forests, thereby maintaining long-term development and economic efficiency. The government can put out the fire by remote monitoring, personnel extinguishing and helicopter extinguishing. Different from most articles on how to eliminate forest fires from the technical point of view, this article mainly analyzes from the two aspects of optimizing resource allocation and guiding policy formulation. This article constructs the differential game model under these three modes, and then obtains the equilibrium result. And the comparative analysis. Finally, the conclusion is drawn. The stronger the wind, the more residents need to flee. However, strong winds are not conducive to crews and helicopters fighting the fire. Rather than fighting fires, residents are more inclined to detect forest fires in time through remote monitoring. When the personnel can effectively control the fire, the personnel fire extinguishing mode is preferentially selected. Otherwise, helicopter firefighting mode should be selected.

## 1. Introduction

### 1.1 Background and research significance

According to FAO's Global Forest Resources Assessment 2020 Key findings, 31 percent of the world's land is covered by forests [1]. Forests can produce oxygen, purify air, conserve water, and conserve water and soil. Forests provide a large number of raw materials such as wood and rubber for people's production and life. Forest ecosystems are also natural habitats for wildlife [2]. It has played a huge role in protecting species diversity. Maintaining the stability of forest ecosystem is of vital significance to human survival and development.

In recent years, the forest has also suffered some destruction. The world's total stock of trees has fallen slightly, from 560 billion cubic meters in 1990 to 557 billion cubic meters in 2020 [1]. Among them, forest fire is an important cause of forest area reduction. Every year

Planning Foundation of Chongqing in China (2021BS080). The funders had no role in study design, data collection and analysis, decision to publish, or preparation of the manuscript.

**Competing interests:** The authors have declared that no competing interests exist.

there will be many forest fires, so that the forest has been seriously damaged. Forest fires can directly reduce forest area by destroying trees. This can seriously damage the forest structure and forest environment [3]. In this case, the forest ecosystem is easily out of balance. Forest fires can lead to a decline in the number and variety of wild plants and animals. If fires are of high intensity, they can disrupt soil chemistry [4], raising the water table in the low depression in the forest, resulting in forest swamps. In addition, it will also cause air pollution, water resources drop, thus affecting people's normal life. Smoke from forest fires can be harmful to people's health, especially children, the elderly and people with respiratory problems. In some cases, forest fires can damage properties and threaten people's lives [5]. Once a forest is destroyed, it is very difficult to restore the original ecosystem.

In order to deal with forest fires, forests need to be managed effectively. For example, forest fires can be managed through mobilization and population cooperation [6]. Compared with developed countries, forest management problems are more prominent in developing countries. Currently, more than 2 billion hectares of forests worldwide have management plans, including most of the forests in Europe. On the other hand, less than 25 percent of forests in Africa and less than 20 percent in South America have management plans [1]. The common forest fire management methods include remote monitoring and timely extinguishing of forest fires. And the forest fire extinguish method is mainly divided into personnel extinguish and helicopter extinguish two kinds. The difficulty of extinguishing forest fires is easily restricted by wind speed, terrain complexity and local conditions.

It is important to choose reasonable and effective forest fire management modes, because not having the right management modes can cause significant harm to the environment, the economy, and human health. If not properly managed, this can lead to environmental impacts such as repetitive burning and land erosion. If the forest fire control mode is too aggressive or inappropriate, it could harm economic interests such as forestry, timber and tourism. However, effective management methods can protect the important ecological functions of forests, thereby sustaining long-term development and economic benefits [7]. Forest fire control is dangerous and can threaten the lives and property of those who control it and the surrounding communities. If a reasonable forest fire control mode is selected, the life and property safety of local residents can be protected to the greatest extent. Reasonable and effective forest fire control modes are very important, and each control method needs to take into account the environmental, economic and human health impacts, and take comprehensive measures to ensure the sustainability and effectiveness of the whole control process [8].

## 1.2 Literature review

The research on forest fire is mainly from the causes of forest fire, the impact of forest fire and how to control forest fire.Some scholars have studied the causes of forest fires. For example, Farinha et al. believes that the more vulnerable the society is, the more likely forest fires will occur [9]. Tan et al. believed that human factors and drought were important causes of forest fires in Indonesia [10]. Wang et al. [11] analyzed forest fire risk areas in Sichuan and believed that climate warming would lead to increased forest fire risk. The scholars studied the main causes of forest fires from social, man-made and weather perspectives.

Some scholars have studied the effects of fire on forest ecosystems. For example, forest fires can affect tree mortality [12]. Years after forest fires, although the number of trees can recover to the original level, tree species will decrease [2]. Multiple fires can lead to vegetation degradation [13]. Forest fires can cause rivers to return to their original condition in between 5 and 45 years [14]. These studies cover the major impacts of forests on the ecological environment.

However, forest fires are constantly changing with terrain and weather conditions, and such changes are not described by the above scholars.

Some scholars have studied how to control forest fires. For example, Vela et al. [15] predicted the spread of forest fires in the Amazon. Vigna et al. [16] argued that the social ecosystem (SES) approach plays an important role in forest fire management. Qiang et al. [17] studied the use of TRPCA and TSVB methods for forest fire monitoring. Vinodhini et al. [18] studied how the MLP and AROC methods based on the internet of things can monitor forest fires. Hu et al. [19] studied the MVMNet method for rapid detection of forest fire smoke. These cover the main methods of forest fire suppression. However, these scholars mainly analyzed how to control forest fires from a technical perspective, rather than a management perspective.

In order to make up for the shortcomings of the above research, this article studies from the perspective of management mode and uses the time-continuous differential game. In this article, the differential game models of remote monitoring, personnel extinguishing and helicopter extinguishing are constructed. The equilibrium results of various models are compared and analyzed. Finally, the applicable scope of three forest fire management modes is obtained. It provides reference for better protection of forest ecosystem.

This article has some contribution to the field of forest fire fighting. The existing research on forest fire fighting is mainly carried out from the angle of improving fire fighting efficiency and improving the development of fire fighting technology and equipment. The analysis of this article is mainly introduced from the optimization of resource allocation and guidance of policy formulation. This study is helpful for the government to formulate perfect fire fighting guidance policies and optimize the allocation of fire fighting resources.

## 2. Methodology

### 2.1 Problem description, hypothesis, and variable definition

**2.1.1 Problem description.** Forest seriously damage forest structure and forest environment, resulting in forest ecosystem out of balance. In order to describe the whole process of forest fire dynamic change, differential game is used in this article. In the process of forest fire control, the remote monitoring of forest fire can detect the fire in time and remind the nearby residents to flee. However, this method is less weather-resilient and does not directly extinguish forest fires. Although the personnel fire extinguishing mode can effectively and accurately extinguish forest fires, it is easy to cause casualties. Helicopters can help local residents escape and reduce casualties, but they are expensive.

Many areas of the earth have forests. The study of forest fire control in this article is applicable to all areas of the earth with forest. In order to protect the ecological environment and people's survival and development needs, it is necessary to carry out forest fire research on a global scale. Specifically, in areas prone to forest fires, such as Southeast Asia, Australia, Africa, North America and other places, it is especially necessary to conduct in-depth research on forest fires. In addition, with the intensification of global climate change, the frequency and scale of forest fires are also increasing, so the research and prevention of forest fires are more urgent, and more investment and control are needed.

In order to effectively control forest fires, the government mainly adopts the following three management modes:

1. Remote monitoring mode. Remote monitoring of forest fires is needed in order to detect ignition points in the forest in time. For this purpose, the government can deploy remote monitoring equipment, and effectively monitor and track hot spot warnings within a

certain range. The main remote monitoring equipment includes these types. First, fire monitors. It uses infrared and thermal imaging technology to detect smoke and heat sources, which can quickly and accurately spot fires. Second, fire warning system. Through the collection and analysis of weather, temperature, humidity, wind speed and other data, it predicts the time and place where the fire may break out, and gives early warning [20]. Third, drones. It can use high-definition cameras for real-time shooting and monitoring of the fire scene, while putting firefighting supplies on the fire scene to mitigate the fire and provide more accurate information for firefighters. Fourth, satellite remote sensing. Satellite imagery can provide information on the extent, intensity and hot spots of fires, which can help fire departments better understand and respond to fires. Fifth, artificial intelligence systems. It uses techniques such as machine learning to analyze large amounts of data and can quickly and accurately identify the occurrence of a fire and its possible scope of impact [21]. The input of remote monitoring equipment and personnel plays a very important role in forest fire early warning. For example, the government determines whether forest fires occur in San Diego by analyzing the temporal brightness and temperature of forest fires [22].

2. Personnel extinguishing mode. In case of forest fire, relevant fire fighters should be assigned tasks. People concerned should work together and use scientific methods to fight forest fires. For example, create fire breaks and use fire extinguishing tools to contain fires. Use water cannons where water is available. When the fire is out, the crew will carry out a blanket inspection of the fire area to see if there are any unextinguished sparks. This should ensure that the fire is completely extinguished. This type of fire fighting is not easy to carry out in the complex terrain. At the same time, it is easy to cause casualties.

3. Helicopter extinguishing mode. Prevention alone cannot completely eliminate the possibility of wildfires, only can reduce the probability of occurrence. Helicopter firefighting needs to take into account factors such as availability, terrain and water sources. If helicopters are well-resourced and easily mobilised, rescue services can quickly dispatch helicopters for rescue missions, reducing delays in rescue operations. At the same time, it can carry out rescue in a wider area, as well as respond to a variety of complex rescue missions. The complexity of the terrain and hard-to-reach areas can limit the choice of landing sites for helicopters. Rugged terrain, narrow valleys, dense forests, etc., can make suitable landing sites scarce. Helicopters need to find a flat, open and safe place to land to ensure the safety of pilots and rescuers. Water source plays a vital role in fire fighting and rescue by helicopter. When there is a forest fire or other fire, the helicopter can use the water to fight the fire from the air. The reliability and proximity of the water source determines whether the helicopter can obtain the water source in time and carry out effective fire fighting operations. The availability and suitability of water sources also affect the duration and effectiveness of fire suppression. Given the availability of helicopters, terrain and water, the most effective way is to send out helicopters to fight the fire. Helicopters can put out fires by dropping water directly on them from the air. This method has the characteristics of quick effect and large water load.

The relationship between the three forest fire control modes is shown in Fig 1.

**2.1.2 Hypothesis.** In order to compare and analyze the application scope of three forest fire management modes, namely remote monitoring, personnel fire fighting and helicopter fire fighting, and to characterize the differential game model of forest fire management, this article establishes the following assumptions based on the game parties, decision-making state and economic development level.

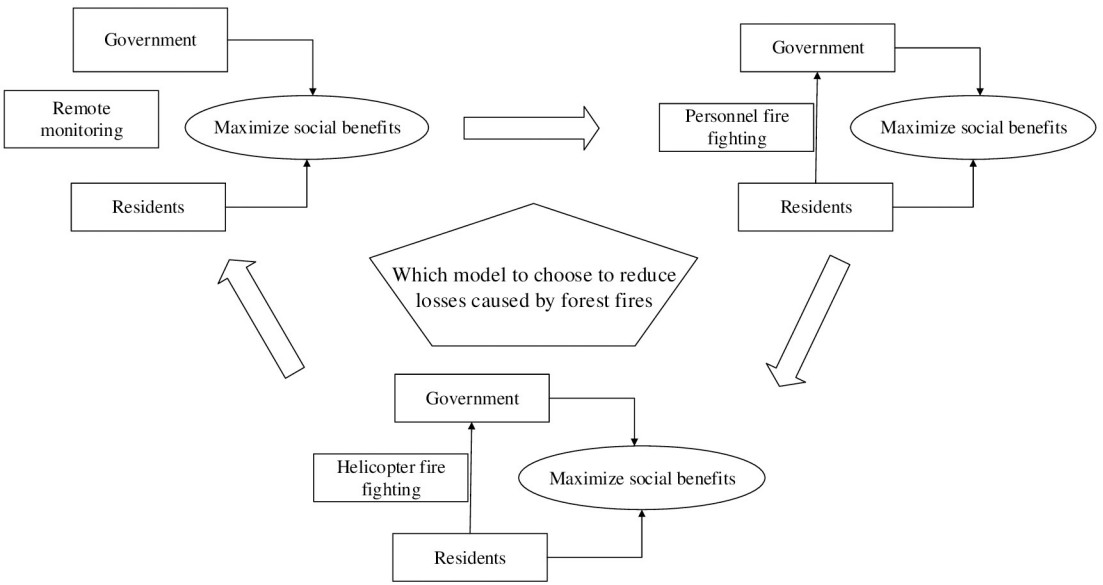

**Fig 1. Relationship between three different forest fire control modes.**

1. There are a large number of residents near the forest. There are more water resources, animals and plants around the forest, and the air is fresh. At the same time, the soil near the forest is good, which can provide soil for people to cultivate. The area around the forest is more suitable for human habitation. For example, the Rocky Mountains are in northern Canada, and there are more forests near them. The forest still has a beautiful natural environment and the ecological environment has not been damaged [23]. For six years in a row since 1994, the United Nations Development Program has ranked the Rocky Mountains as the best place in the world to live. Of course, not all forests are suitable for human habitation. For example, the Amazon rainforest is too humid for human habitation. For simplicity, uninhabitable forests are not the subject of this article.

2. The decision-making of the government and residents is constantly changing. When forest fires occur, the government will make corresponding decisions, such as dispatching fire fighters or helicopters for rescue [24]. And the government's decisions are influenced by weather conditions or how stranded residents are. The government's fire-fighting measures will also affect the movement of residents to flee. When the government does a better job of controlling fires, residents' fears are mitigated. If the government doesn't act, or if the fire gets out of hand, it could affect how much people flee. At the same time, the level of flight of the local population in turn has a very important influence on the government's decision-making. As a result, the decisions of the government and residents are constantly changing.

3. The country has a relatively high level of economic development. If a country has a low level of economic development, the government will focus more on feeding its own people. The government has no extra money to introduce remote surveillance systems. When there is a forest fire, the government does not have the money to hire firefighters to fight it. In such cases, governments do not have the money to fight fires or rescue, even if they know they can get more bang for their buck. In reality, more than half of the world's forests (54%) are found in just five countries: the Russian Federation, Brazil, Canada, the United States of America and China [1]. These countries have a higher level of economic development and

are able to organize human and material resources to deal with forest fires. Therefore, for the sake of convenience, this article assumes that a country has a high level of economic development. When there is a forest fire, there is enough money to send out personnel or helicopters.

**2.1.3 Variable definition.** When constructing the differential game model in this article, many parameters and variables are designed. These parameters and variables are defined as shown in Table 1.

Unit personnel input $b_{M1}$ to eliminate forest fires can produce many benefits, including the following aspects. First, protect the safety of life and property. Forest fires threaten people's life safety and property interests. The input of unit personnel can effectively contain and

**Table 1. The main definition of variables and parameters in this article.**

| variables and parameters | specific meaning |
|---|---|
| $Y = \{R,P,H\}$ | three forest fire management modes (remote monitoring, personnel fire fighting, helicopter fire fighting) |
| independent variable | |
| $M_{Yi}(t)$ | government personnel input under forest fire control mode $Y$ |
| $T_{Yi}(t)$ | government equipment input under forest fire control mode $Y$ |
| $E_{Yi}(t)$ | the escape degree of residents to forest fires under the forest fire management mode $Y$ |
| $x_{Y1}(t)$ | the reputation of the government under the forest fire management model $Y$ |
| $x_{Y2}(t)$ | residents' satisfaction under forest fire management mode $Y$ |
| parameter | |
| $\rho$ | the discount rate occurring over time, $0 \leq \rho \leq 1$ |
| $\delta$ | the rate of decay of government reputation or public satisfaction, $\delta > 0$ |
| $b_{M1}$ | income generated by unit personnel input, $b_{M1} > 0$ |
| $b_{T1}$ | the effect of a unit of equipment input, $b_{T1} > 0$ |
| $c_{T1}$ | the cost per unit of equipment, $c_{T1} > 0$ |
| $c_{M1}$ | the cost per unit of personnel, $c_{M1} > 0$ |
| $a_1$ | the impact of equipment or personnel input on reputation, $a_1 > 0$ |
| $a_2$ | the effect of escaping a fire on satisfaction, $a_2 > 0$ |
| $b_E$ | the income earned by unit residents fleeing, $b_E > 0$ |
| $b_{T1}$ | revenue per unit of helicopter deployed to fight a fire, $b_{T1} > 0$ |
| $v_w$ | wind speed at the time of the forest fire, $v_w > 0$ |
| $w_s$ | degree of weather stability, $w_s > 0$ |
| $v_s$ | the positive impact of remote monitoring equipment or technology on personnel fighting forest fires, $v_s > 0$ |
| $d$ | the distance of the helicopter from the fire, $d > 0$ |
| $d_{R2}, d_{P2}, d_{H2}$ | the distance of the ignition from the residence, $d_{R2}, d_{P2}, d_{H2} > 0$ |
| $l$ | the positive effect of government reputation or public satisfaction on earnings, $l > 0$ |
| $I_{R2}$ | fire information for residents, $I_{R2} > 0$ |
| $p_{H2}$ | the positive effect of helicopter fire fighting on the evacuation of residents, $p_{H2} > 0$ |
| $q_{P1}$ | the proportion of casualties caused by personnel fighting forest fires, $0 < q_{P1} < 1$ |
| function | |
| $J_{Y1}(t)$ | social welfare function of government under forest fire control mode $Y$ |
| $J_{Y2}(t)$ | social welfare function of residents under forest fire control mode $Y$ |
| $V_{Y1}(t)$ | government's benefit function under forest fire control mode $Y$ |
| $V_{Y2}(t)$ | benefit function of residents under forest fire control mode $Y$ |

extinguish forest fires, and protect people's life safety and property from fire. This will reduce casualties and property losses, and bring huge benefits to society. Second, protect the ecological environment. Forest is one of the important ecosystems on the earth, forest fires will not only destroy vegetation and soil, but also release a lot of carbon emissions, causing serious impact on the environment. The input of unit personnel can prevent the spread of fire, reduce the scope of forest destruction, and protect the stability and health of the ecological environment. Third, maintain social and economic stability. The outbreak and spread of forest fires will not only cause direct economic losses to surrounding residents and farmers, but also have a huge impact on tourism, agriculture, forestry and other related industries. The input of unit personnel can quickly respond to fires, effectively control and extinguish fires, and maintain social and economic stability. By preventing and extinguishing forest fires, the rational utilization of personnel and financial resources can be realized, the loss of personnel and property can be reduced to the greatest extent, the ecological environment can be protected, and social stability can be maintained. These are important gains that have profound implications for individuals, communities and society as a whole.

It is worth noting that the wind speed $v_w$ will have an impact on the social benefits of the government and residents. Stronger winds can accelerate the spread of fires because they provide more oxygen and carry flames and smoke farther away. This will make the fire more intense, burn over a larger area and for longer. In addition, strong winds can spread the fire downhill. This will threaten people's property and life safety and discourage people from fleeing.

When forest fires occur in areas with complex terrain such as mountains or hills, there may be steep hillsides, canyons, rivers and other obstacles at the fire site, which is difficult for ground rescue personnel to enter and operate. At this time, the helicopter can show its skills and quickly reach the designated area to help the trapped people escape the fire smoothly, thus having a positive impact $p_{H2}$ on the rescue.

The social benefits of forest fire control mainly include life safety, ecological protection, property protection, fresh air, productivity protection and so on. This is because forest fire control through remote monitoring, personnel fire fighting, helicopter fire fighting can reduce the harm and threat of fire to human beings. The risk of injury and death can be reduced by early detection and control of fires. Forest is one of the important ecosystems on earth. Eliminating forest fires is conducive to maintaining ecological balance, reducing the death of vegetation and wildlife, and reducing the destruction of ecosystem. Forest fires will cause great losses to people's property, such as forest burning, wildlife loss, farmland and residential houses are burned down, etc. Eliminating forest fires can avoid the above economic losses. Forest fires smoke all over the sky, and a sharp decline in air quality will have a greater impact on health. Controlling and eliminating forest fires can reduce the concentration of pollutants in the air and protect people's health. The elimination or early control of forest fires can avoid the loss of productivity caused by forest fires, such as forest logging, fishing, tourism and other related industries.

The evaluation of social benefits can start from these aspects. The safety of life benefits of forest fire control can be evaluated by statistical analysis of the number of people rescued by the fire department, the fatality rate of the fire and the medical costs. To assess the ecological benefits of forest fire control, data can be collected and analyzed in terms of forest area, forest destruction, plant recovery, and wildlife loss. The property protection benefits of forest fire control can be analyzed in terms of the reduction of economic losses caused by fire, the loss of property of enterprises and residents, and the reduction of insurance costs. Assessing the clean air benefits of controlling forest fires can be derived from air quality monitoring data, such as particulate matter concentrations, as well as from people's physical health. The productivity

protection benefits of forest fire control can be assessed by statistics and analysis in terms of the earnings of related industries, asset values, employment opportunities and business operations.

## 2.2 Differential game of different forest fire control modes

Differential game refers to a time continuous game played by multiple players in a time continuous system. It has the goal of optimizing the independence and conflict of each player, and can finally obtain the strategy of each player evolving over time and reach the Nash equilibrium. At present, the differential game it is mainly applied in the fields of logistics management [25], advertising decision [26], supply chain [27], etc. The situation of forest fires varies according to weather conditions. The level of the government s response varies with the situation of the terrain, equipment and personnel. In order to describe this change clearly, this article uses the time continuous differential game method.

**2.2.1 Remote monitoring.** In the mode of remote monitoring of forest fires, the main work that the government can do is to monitor forest fires through advanced technology. At this point, the utility of the government is related to the number of personnel and equipment invested in remote monitoring. Meanwhile, the effectiveness of the government's remote monitoring is also limited by the stability of the weather. In the event of a forest fire, the government can provide fire-related information to local residents, so as to help local residents to escape.

In the remote monitoring mode, the social benefits obtained by the government and residents are:

$$J_{R1} = \int_0^\infty \left[ b_{M1} M_{R1}(t) \ln(1 + w_S) - \frac{c_{M1}}{2} M_{R1}^2(t) + b_{T1} T_{R1}(t) - \frac{c_{T1}}{2} T_{R1}^{\frac{1}{2}}(t) + lx_{R1}(t) \right] e^{-\rho t} dt \quad (1)$$

$$J_{R2} = \int_0^\infty \left[ -\frac{v_w d_{R2}}{\ln(e + I_{R2})} E_{R2}^{\frac{1}{2}}(t) + b_E E_{R2}(t) + lx_{R2}(t) \right] e^{-\rho t} dt \quad (2)$$

In the above formula, $b_{M1} M_{R1}(t) \ln(1 + w_S)$ represents the effect of remote monitoring on forest fire prevention. $\ln(1 + w_S)$ represents the positive effect of weather stability on forest fire prevention. $\frac{c_{M1}}{2} M_{R1}^2(t)$ represents the personnel cost invested in remote monitoring mode. $b_{T1} T_{R1}(t)$ represents the income generated by investing related monitoring equipment in remote monitoring mode. $\frac{c_{T1}}{2} T_{R1}^{\frac{1}{2}}(t)$ represents the cost generated by investing related monitoring equipment in remote monitoring mode. $lx_{R1}(t)$ represents the reputation gained by the government under the mode of remote monitoring.

$b_E E_{R2}(t)$ represents the income earned by residents for fleeing the fire. $\ln(e + I_{R2})$ represents the positive impact of bushfire-related information received by residents on escape from the fire. $\frac{v_w d_{R2}}{\ln(e + I_{R2})} E_{R2}^{\frac{1}{2}}(t)$ represents the cost to residents of the wildfires. There is a reason for this. In the event of a fire, the greater the wind speed, the greater the speed and extent of the fire, and the faster the smoke will spread over the surrounding area, making the whole situation more complex and dangerous. Due to the shielding and interference of smoke and dust, it may bring many unfavorable factors to the escape of residents. When the wind speed is large, the escape path of residents may be cut off by fire and smoke or be intensified and narrowed. This will increase the difficulties and hidden dangers of escape. So, the wind speed is proportional to the cost of fleeing. $lx_{R2}(t)$ represents the residents' satisfaction with forest fire prevention.

In the remote monitoring mode, the change of government reputation is:

$$\dot{x}_{R1}(t) = a_1(M_{R1}(t) + T_{R1}(t)) - \delta x_{R1}(t) \tag{3}$$

In the remote monitoring mode, the change of public satisfaction is:

$$\dot{x}_{R2}(t) = a_2 E_{R2}(t)\ln(1 + I_{R2}) - \delta x_{R2}(t) \tag{4}$$

In the above formula, $a_1(M_{R1}(t)+T_{R1}(t))$ represents an increased reputation for government investment in remote monitoring personnel and equipment. $a_1 M_{R1}(t)$ represents the government's increased reputation for spending on remote surveillance personnel. $a_1 T_{R1}(t)$ represents the increased prestige of the government's investment in remote surveillance equipment. $\delta x_{R1}(t)$ represents a decline in the government's reputation. $a_2 E_{R2}(t)\ln(1+I_{R2})$ represents the satisfaction of the public from remotely monitored escape from a forest fire. $\delta x_{R2}(t)$ represents the decline in public satisfaction.

**2.2.2 Personnel fire fighting.** In the personnel fire fighting mode, the social benefits obtained by the government and residents are:

$$J_{P1} = \int_0^\infty \left[ b_{M1}M_{P1}(t) - v_w q_{P1}M_{P1}(t) - \frac{c_{M1}}{2}M_{P1}^2(t) + lx_{P1}(t) \right]e^{-\rho t}dt \tag{5}$$

$$J_{P2} = \int_0^\infty \left[ -v_w d_{P2}E_{P2}^{\frac{1}{2}}(t) + b_E E_{P2}(t) + lx_{P2}(t) \right]e^{-\rho t}dt \tag{6}$$

In the above formula, $b_{M1}M_{P1}(t)$ represents the benefit of personnel fighting fire. $v_w q_{P1}M_{P1}(t)$ represents the risk of loss of life in the mode of personnel fighting forest fires. $\frac{c_{M1}}{2}M_{P1}^2(t)$ represents the personnel cost invested in the mode of personnel extinguishing fire. $lx_{P1}(t)$ represents personnel put out the fire under the model, the government gained a reputation.

$b_E E_{P2}(t)$ represents the income earned by residents for fleeing the fire. $v_w d_{P2}E_{P2}^{\frac{1}{2}}(t)$ represents the cost to residents of a forest fire in the mode of personnel fighting the fire. $lx_{P2}(t)$ represents the residents' satisfaction with forest fire prevention.

In the personnel fire fighting mode, the change of government reputation is:

$$\dot{x}_{P1}(t) = a_1 M_{P1}(t) - \delta x_{P1}(t) \tag{7}$$

In the personnel fire fighting mode, the change of public satisfaction is:

$$\dot{x}_{P2}(t) = a_2 E_{P2}(t) - \delta x_{P2}(t) \tag{8}$$

In the above formula, $a_1 M_{P1}(t)$ represents the increased prestige of the government's commitment to firefighters. $\delta x_{P1}(t)$ represents a decline in the government's reputation. $a_2 E_{P2}(t)$ represents the public satisfaction that comes from escaping a forest fire. $\delta x_{P2}(t)$ represents a decline in public satisfaction.

### 2.2.3 Helicopter fire fighting

In the helicopter fire fighting mode, the social benefits obtained by the government and residents are:

$$J_{H1} = \int_0^\infty \left[ b_{T1} T_{H1}(t) - dv_w T_{H1}^2(t) + l x_{H1}(t) \right] e^{-\rho t} dt \tag{9}$$

$$J_{H2} = \int_0^\infty \left[ -v_w d_{H2} E_{H2}^{\frac{1}{2}}(t) + b_E E_{H2}(t) \ln(e + p_{H2}) + l x_{H2}(t) \right] e^{-\rho t} dt \tag{10}$$

In the above formula, $b_{T1} T_{H1}(t)$ represents the benefit of helicopter fighting forest fires. $dv_w T_{H1}^2(t)$ represents the cost of equipment invested in the mode of helicopter fire extinguishing. $l x_{H1}(t)$ represents the helicopter fire fighting model under which the government gained a reputation.

$b_E E_{H2}(t) \ln(e+p_{H2})$ represents the income earned by residents for fleeing the fire. $\ln(e+p_{H2})$ represents the helicopter helped residents escape the fire. This is because helicopters can quickly transport the injured who need more rescue, and for many residents, they can escape the fire without relying on helicopters. The effect of helicopters on the flight of residents was first to increase rapidly and then to increase slowly. It is in the form of a pairwise function. $v_w d_{H2} E_{H2}^{\frac{1}{2}}(t)$ represents the cost to residents of a bushfire in helicopter model. $l x_{H2}(t)$ represents the residents' satisfaction with forest fire prevention.

In the helicopter fire fighting model, the change of government reputation is:

$$\dot{x}_{H1}(t) = a_1 T_{H1}(t) - \delta x_{H1}(t) \tag{11}$$

In the helicopter fire fighting model, the change of public satisfaction is:

$$\dot{x}_{H2}(t) = a_2 E_{H2}(t) - \delta x_{H2}(t) \tag{12}$$

In the above formula, $a_1 M_{H1}(t)$ represents the government has increased its reputation by investing in helicopters. $\delta x_{H1}(t)$ represents a decline in the government's reputation. $a_2 E_{H2}(t)$ represents the public satisfaction that comes from escaping a forest fire. $\delta x_{H2}(t)$ represents a decline in public satisfaction.

## 3. Results

In the differential game, the social welfare of the government and residents when forest fires occur is not only affected by the control variables and parameters, but also constantly changes with the influence of time, state and country on social welfare. In order to better calculate the input of government equipment, the input of personnel, the fleeing degree of residents, and the social benefits of the government and residents, the HJB formula is adopted. HJB formula is a partial differential equation, which is the core of optimal control.

### 3.1 HJB formula

If the government uses remote monitoring equipment to control forest fires, the HJB equations of the social welfare function of the government and residents are:

$$
\begin{aligned}
\rho V_{R1} &= \max_{M_{R1}(t), T_{R1}(t)} \left\{ \left[ b_{M1} M_{R1}(t) \ln(1 + w_S) - \frac{c_{M1}}{2} M_{R1}^2(t) + b_{T1} T_{R1}(t) - \frac{c_{T1}}{2} T_{R1}^{\frac{1}{2}}(t) + lx_{R1}(t) \right] \right. \\
&\quad \left. + \frac{\partial V_{R1}}{\partial x_{R1}} \left[ a_1 (M_{R1}(t) + T_{R1}(t)) - \delta x_{R1}(t) \right] \right\}
\end{aligned}
\tag{13}
$$

$$
\rho V_{R2} = \max_{E_{R2}(t)} \left\{ \left[ -\frac{v_w d_{R2}}{\ln(e + I_{R2})} E_{R2}^{\frac{1}{2}}(t) + b_E E_{R2}(t) + lx_{R2}(t) \right] + \frac{\partial V_{R2}}{\partial x_{R2}} \left[ a_2 E_{R2}(t) \ln(1 + I_{R2}) - \delta x_{R2}(t) \right] \right\}
\tag{14}
$$

If the government uses personnel fire fighting model to control forest fires, the HJB equations of the social welfare function of the government and residents are:

$$
\rho V_{P1} = \max_{M_{P1}(t)} \left\{ \left[ b_{M1} M_{P1}(t) - v_w q_{P1} M_{P1}(t) - \frac{c_{M1}}{2} M_{P1}^2(t) + lx_{P1}(t) \right] + \frac{\partial V_{P1}}{\partial x_{P1}} \left[ a_1 M_{P1}(t) - \delta x_{P1}(t) \right] \right\}
\tag{15}
$$

$$
\rho V_{P2} = \max_{E_{P2}(t)} \left\{ \left[ -v_w d_{P2} E_{P2}^{\frac{1}{2}}(t) + b_E E_{P2}(t) + lx_{P2}(t) \right] + \frac{\partial V_{P2}}{\partial x_{P2}} \left[ a_2 E_{P2}(t) - \delta x_{P2}(t) \right] \right\}
\tag{16}
$$

If the government uses helicopter fire fighting model to control forest fires, the HJB equations of the social welfare function of the government and residents are:

$$
\rho V_{H1} = \max_{T_{H1}(t)} \left\{ \left[ b_{T1} T_{H1}(t) - dv_w T_{H1}^2(t) + lx_{H1}(t) \right] + \frac{\partial V_{H1}}{\partial x_{H1}} \left[ a_1 T_{H1}(t) - \delta x_{H1}(t) \right] \right\}
\tag{17}
$$

$$
\rho V_{H2} = \max_{E_{H2}(t)} \left\{ \left[ -v_w d_{H2} E_{H2}^{\frac{1}{2}}(t) + b_E E_{H2}(t) \ln(e + p_{H2}) + lx_{H2}(t) \right] + \frac{\partial V_{H2}}{\partial x_{H2}} \left[ a_2 E_{H2}(t) - \delta x_{H2}(t) \right] \right\}
\tag{18}
$$

### 3.2 Result of equilibrium

Proposition 1: In the remote monitoring model, input of government equipment, the input of personnel, the fleeing degree of residents, and the social benefits of the government and residents are respectively (See Appendix 1 in S1 File for details):

$$
M_{R1}^*(t) = \frac{b_{M1} \ln(1 + w_S) + a_1 \frac{l}{\rho + \delta}}{c_{M1}}
\tag{19}
$$

$$T_{R1}^*(t) = \frac{c_{T1}^2}{16\left(b_{T1} + a_1 \frac{l}{\rho+\delta}\right)^2} \tag{20}$$

$$E_{R2}^*(t) = \frac{v_w^2 d_{R2}^2}{\ln^2(e + I_{R2})\left[2b_E + 2\frac{l}{\rho+\delta}a_2 \ln\left(1 + I_{R2}\right)\right]^2} \tag{21}$$

$$
\begin{aligned}
V_{R1}^* &= \frac{l}{\rho+\delta}x_{R1} + \frac{1}{\rho}b_{M1}\frac{b_{M1}\ln\left(1 + w_s\right) + a_1\dfrac{l}{\rho+\delta}}{c_{M1}}\ln(1 + w_s) - \frac{1}{\rho}\frac{c_{M1}}{2}\left[\frac{b_{M1}\ln\left(1 + w_s\right) + a_1\dfrac{l}{\rho+\delta}}{c_{M1}}\right]^2 \\
&+ \frac{1}{\rho}b_{T1}\frac{c_{T1}^2}{16\left(b_{T1} + a_1\dfrac{l}{\rho+\delta}\right)^2} - \frac{c_{T1}}{2}\frac{c_{T1}}{4\left(b_{T1} + a_1\dfrac{l}{\rho+\delta}\right)} \\
&+ \frac{1}{\rho}\frac{l}{\rho+\delta}a_1\left(\frac{b_{M1}\ln(1 + w_s) + a_1\dfrac{l}{\rho+\delta}}{c_{M1}} + \frac{c_{T1}^2}{16\left(b_{T1} + a_1\dfrac{l}{\rho+\delta}\right)^2}\right)
\end{aligned} \tag{22}
$$

$$
\begin{aligned}
V_{R2}^* &= \frac{l}{\rho+\delta}x_{R2} - \frac{1}{\rho}\frac{v_w d_{R2}}{\ln(e + I_{R2})}\frac{v_w d_{R2}}{\ln(e + I_{R2})\left[2b_E + 2\dfrac{l}{\rho+\delta}a_2\ln\left(1 + I_{R2}\right)\right]} \\
&+ \frac{1}{\rho}\frac{b_E v_w^2 d_{R2}^2}{\ln^2(e + I_{R2})\left[2b_E + 2\dfrac{l}{\rho+\delta}a_2\ln\left(1 + I_{R2}\right)\right]^2} \\
&+ \frac{1}{\rho}\frac{l}{\rho+\delta}a_2\frac{v_w^2 d_{R2}^2}{\ln^2(e + I_{R2})\left[2b_E + 2\dfrac{l}{\rho+\delta}a_2\ln\left(1 + I_{R2}\right)\right]^2}\ln(1 + I_{R2})
\end{aligned} \tag{23}
$$

Conclusion 1: In remote monitoring mode, the more stable the weather is, the more personnel need to be deployed to monitor the fire. The greater the cost of the equipment, the more necessary to invest the equipment for remote monitoring. And the better the effect of unit number of equipment, the more need to reduce the input of this equipment. The more information residents had about the fire, the less they fled. The stronger the wind speed at the time of the fire, the more residents fled.

Proposition 2: In the personnel fire fighting model, input of government equipment, the fleeing degree of residents, and the social benefits of the government and residents are

respectively (See Appendix 2 in S1 File for details):

$$M_{P1}^*(t) = \frac{b_{M1} - v_w q_{P1} + \frac{l}{\rho+\delta} a_1}{c_{M1}} \tag{24}$$

$$E_{P2}^*(t) = \frac{v_w^2 d_{P2}^2}{4\left(b_E + \frac{l}{\rho+\delta} a_2\right)^2} \tag{25}$$

$$V_{P1}^* = \frac{l}{\rho+\delta} x_{P1} + \frac{1}{\rho} b_{M1} \frac{b_{M1} - v_w q_{P1} + \frac{l}{\rho+\delta} a_1}{c_{M1}} - \frac{1}{\rho} v_w q_{P1} \frac{b_{M1} - v_w q_{P1} + \frac{l}{\rho+\delta} a_1}{c_{M1}}$$
$$- \frac{1}{\rho} \frac{c_{M1}}{2} \left(\frac{b_{M1} - v_w q_{P1} + \frac{l}{\rho+\delta} a_1}{c_{M1}}\right)^2 + \frac{1}{\rho} \frac{l}{\rho+\delta} a_1 \frac{b_{M1} - v_w q_{P1} + \frac{l}{\rho+\delta} a_1}{c_{M1}} \tag{26}$$

$$V_{P2}^* = \frac{l}{\rho+\delta} x_{P2} - \frac{1}{\rho} v_w d_{P2} \frac{v_w d_{P2}}{2\left(b_E + \frac{l}{\rho+\delta} a_2\right)} + b_E \frac{1}{\rho} \frac{v_w^2 d_{P2}^2}{4\left(b_E + \frac{l}{\rho+\delta} a_2\right)^2}$$
$$+ \frac{1}{\rho} \frac{l}{\rho+\delta} a_2 \frac{v_w^2 d_{P2}^2}{4\left(b_E + \frac{l}{\rho+\delta} a_2\right)^2} \tag{27}$$

Conclusion 2: In the personnel fire extinguishing mode, the more income generated by personnel input per unit number, the more need to increase personnel input. The greater the wind speed, the more need to reduce personnel input. The greater the proportion of casualties, the greater the need to reduce personnel input. The stronger the wind, the greater the need for evacuation. The farther away the fire is from the residents, the greater the degree of escape from the fire.

Proposition 3: In the helicopter fire fighting model, input of government equipment, the fleeing degree of residents, and the social benefits of the government and residents are respectively (See Appendix 3 in S1 File for details):

$$T_{H1}^*(t) = \frac{b_{T1} + \frac{l}{\rho+\delta} a_1}{2 d v_w} \tag{28}$$

$$E_{H2}^*(t) = \frac{v_w^2 d_{H2}^2}{4\left[b_E \ln(1 + P_{H2}) + \frac{l}{\rho+\delta} a_2\right]^2} \tag{29}$$

$$V_{H1}^* = \frac{l}{\rho+\delta}x_{H1} + \frac{1}{\rho}b_{T1}\frac{b_{T1}+\frac{l}{\rho+\delta}a_1}{2dv_w} - \frac{1}{\rho}dv_w\left(\frac{b_{T1}+\frac{l}{\rho+\delta}a_1}{2dv_w}\right)^2 + \frac{1}{\rho}\frac{l}{\rho+\delta}a_1\frac{b_{T1}+\frac{l}{\rho+\delta}a_1}{2dv_w} \quad (30)$$

$$V_{H2}^* = \frac{l}{\rho+\delta}x_{H2} - \frac{1}{\rho}v_w d_{H2}\frac{v_w d_{H2}}{2\left[b_E\ln(1+P_{H2})+\frac{l}{\rho+\delta}a_2\right]} + \frac{1}{\rho}b_E\frac{v_w^2 d_{H2}^2}{4\left[b_E\ln(1+P_{H2})+\frac{l}{\rho+\delta}a_2\right]^2}$$

$$\ln(e+p_{H2}) + \frac{1}{\rho}\frac{l}{\rho+\delta}a_2\frac{v_w^2 d_{H2}^2}{4\left[b_E\ln(1+P_{H2})+\frac{l}{\rho+\delta}a_2\right]^2} \quad (31)$$

Conclusion 3: In the mode of forest fire control by helicopter, the input of helicopter is inversely proportional to the wind speed. The amount of helicopter input is inversely proportional to the distance from the fire. Residents fled in direct proportion to the wind speed. The degree to which people flee is directly proportional to the distance from the fire.

## 3.3 Numerical analysis

The discount factor $\rho$ is 0.9. The decay rate $\delta$ of government reputation or public satisfaction is 0.1. The casualty ratio $q_{P1}$ of personnel fighting forest fires is 0.2. The positive effect $p_{H2}$ of helicopter fire fighting on residents' escape is 2. The fire information $I_{R2}$ of residents received is 1.5. The positive impact $l$ of government reputation or public satisfaction on revenue is 1. The distance of the helicopter from the fire is 4. The positive impact $v_s$ of remote monitoring equipment or technology on personnel fighting forest fires is 2. The income $b_E$ from the degree of resident unit flight is 3. The cost $c_{M1}$ per unit of personnel is 3. The cost $c_{T1}$ per unit of equipment is 2. The effect $b_{T1}$ for each unit of equipment input is 2.5. The degree $w_s$ of weather stability is 2. The influence $a_1$ of equipment or personnel input on reputation is 1. The effect $a_2$ of escaping the fire on satisfaction is 1.5.

When the wind speed $v_w$ at the time of the forest fire is 1, this article can calculate:

$$V_{R1}^* = 0.22b_{M1}^2 + 0.41b_{M1} + 1.12 \quad (32)$$

$$V_{P1}^* = 0.184b_{M1}^2 + 0.3b_{M1} + 1.12 \quad (33)$$

$$V_{H1}^* = 1.85 \quad (34)$$

The following graph (named Fig 2) can also be produced.

When the wind speed $v_w$ at the time of the forest fire is 2, this article can calculate:

$$V_{R1}^* = 0.22b_{M1}^2 + 0.41b_{M1} + 1.12 \quad (35)$$

$$V_{P1}^* = 0.184b_{M1}^2 + 0.22b_{M1} + 1.07 \quad (36)$$

$$V_{H1}^* = 1.42 \quad (37)$$

The following graph (named Fig 3) can also be produced.

Conclusion 4: The wind speed is inversely proportional to the social benefits obtained by the government under the personnel fire fighting mode and helicopter fire fighting mode. When the income generated by the personnel input per unit number is small, the social benefit

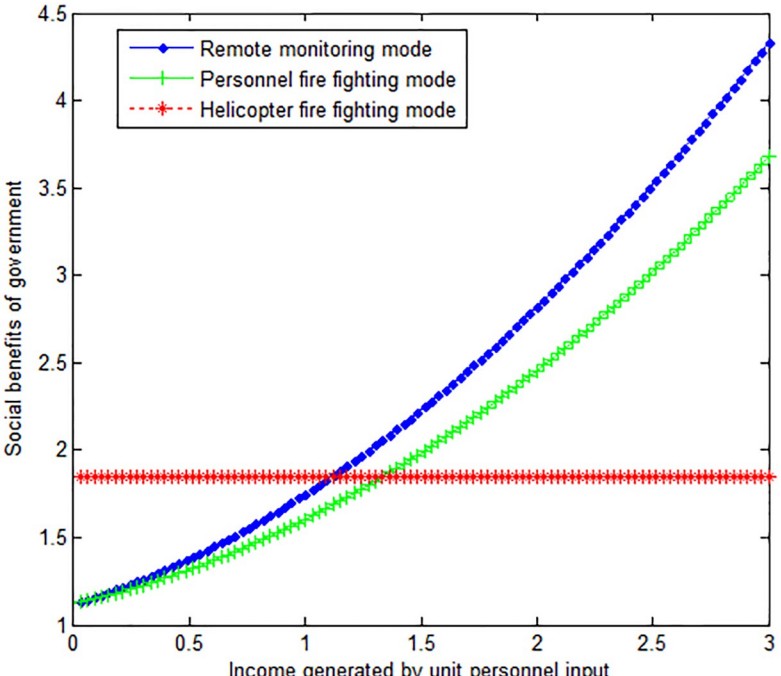

**Fig 2. Impact of income generated by personnel input on the government's social benefits.**

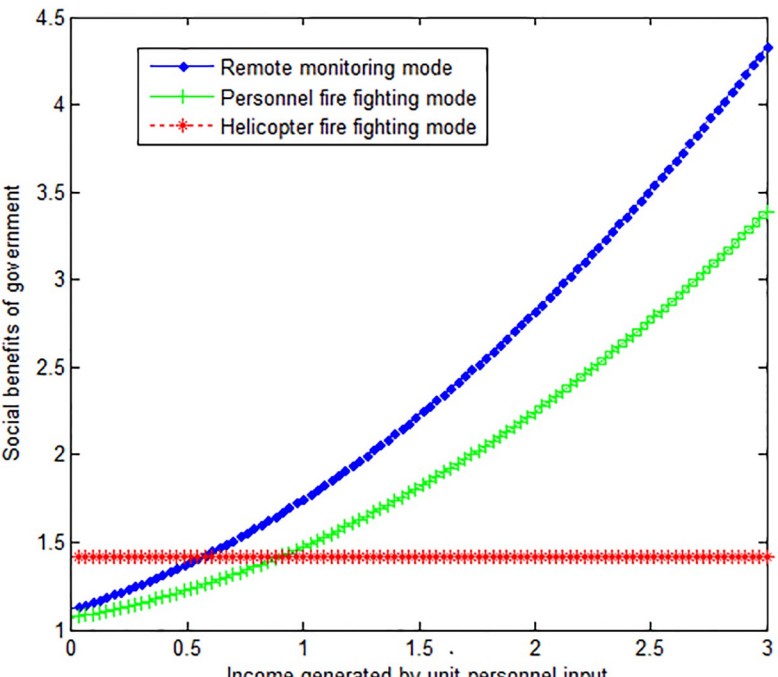

**Fig 3. Impact of income generated by personnel input on the government's social benefits.**

of the government under the helicopter fire-fighting mode is greater than that under the personnel fire-fighting mode and the remote monitoring mode. With the increase of income generated by personnel input per unit number, the social income of the government under personnel fire-fighting mode and remote monitoring mode is increasing. In the end, the social benefit of the government under these two modes is greater than that under the helicopter fire-fighting mode.

When the wind speed $v_w$ at the time of the forest fire is 1, this article can calculate:

$$V_{R2}^* = 1 - 0.031 d_{R2}^2 \tag{38}$$

$$V_{P2}^* = 1 - 0.061 d_{P2}^2 \tag{39}$$

$$V_{H2}^* = 1 - 0.042 d_{H2}^2 \tag{40}$$

The following graph (named Fig 4) can also be produced.

When the wind speed $v_w$ at the time of the forest fire is 2, this article can calculate:

$$V_{R2}^* = 1 - 0.124 d_{R2}^2 \tag{41}$$

$$V_{P2}^* = 1 - 0.244 d_{P2}^2 \tag{42}$$

$$V_{H2}^* = 1 - 0.168 d_{H2}^2 \tag{43}$$

The following graph (named Fig 5) can also be produced.

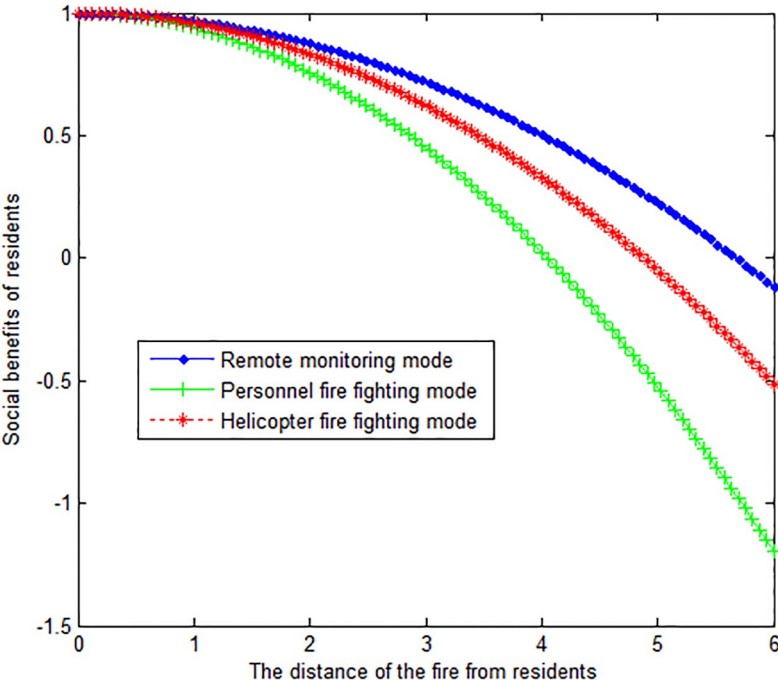

**Fig 4. The effect of fire distance from residents on residents' social benefits.**

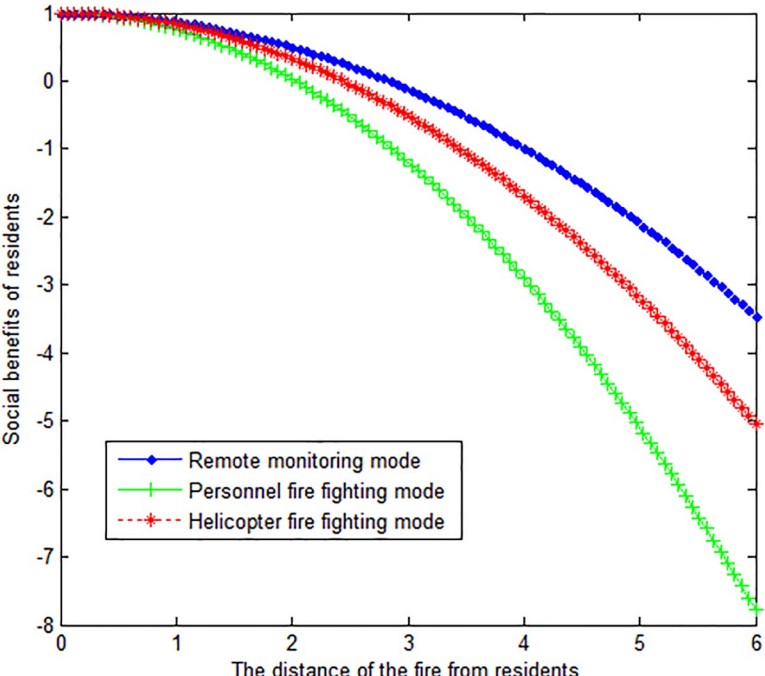

**Fig 5. The effect of fire distance from residents on residents' social benefits.**

Conclusion 5: Under the three fire extinguishing modes, wind speed is inversely proportional to the social benefits of residents. Under the remote monitoring mode, the residents get the greatest social benefits. The social benefits of helicopter firefighting are in the middle. The social benefits of residents are the least under the personnel fire fighting mode.

## 4. Discussion

Because forest fires can spread quickly with strong winds, the government needs to put them out in time. Remote monitoring of forest fires can detect fires in time and alert nearby residents to flee. However, this method is less weather-resilient and does not directly extinguish forest fires. Although the personnel fire extinguishing mode can effectively and accurately extinguish forest fires, it is easy to cause casualties. Helicopters can help local residents escape and reduce casualties, but they are expensive. Therefore, the extinguishment mode of various forest fires is an important issue in this article. Existing studies mainly analyze the effects of each model through relevant data. There is no comparative analysis of the applicable scope of different models. In this article, the differential game is applied in the field of forest fire control, especially considering how to effectively control forest fire in the case of wind speed and fire diffusion.

Forest fires have become more frequent in recent years. The forest area is huge, the terrain is complex and the weather is unstable. Forest fires are hard to spot in time. In the past forest fire monitoring, the fire warning method based solely on video image or thermal imager is very ineffective. In order to detect forest fires as soon as possible, forest fires can be monitored remotely using advanced long-range multispectral scanning. Governments can also use advanced drone surveillance systems [28]. Costly forest fire equipment often achieves better results. This allows forest fires to be detected more promptly. This is similar to the conclusion

Grammalidis [29] reached. Grammalidis [29] believes that more advanced Cubesats have significant advantages over traditional satellites for smoke and fire detection.

Many forest fires are caused by the disposal of combustible materials. When the weather is more stable, it is easier for people to access the forest for productive activities. Therefore, the more stable the weather, the more necessary the relevant personnel to strengthen the monitoring of forest fires. Once the remote monitoring system detects a forest fire, it needs to provide more fire information to the residents in the area where the forest fire occurs. This is to prevent panic among residents. At the same time, increasing the scope of fire monitoring is very important to ensure the safety of people's lives and property. Fire is a kind of destructive disaster, which can spread quickly and cause huge casualties and property losses. Expanding the scope of fire monitoring can detect and alarm fires in advance so that timely measures can be taken for fire fighting and rescue. This is consistent with the expansion of fire monitoring networks proposed by Kumar et al. [30]. The remote monitoring system of forest fires should monitor wind speed and other weather conditions in time. If the wind speed is fast, it is necessary to remind local residents to evacuate quickly and to evacuate further away.

When the wind speed is high, both personnel and helicopters should be reduced. Because high winds can cause fires to spread faster and cause casualties. And high winds are making it harder for helicopters to fight the fires. This is because high wind speed will have an impact on the helicopter's handling ability and stability, which may lead to hovering difficulties, flight instability, and increase the possibility of accidents. Therefore, in extreme wind conditions, the use of fire fighting helicopters is reduced to ensure the safety and effectiveness of flight operations. And Martini et al. [31] came to a similar conclusion in the reconstruction of perturbations on autonomous helicopters. The difference is that when the wind speed is larger, the personnel should stop participating in the forest fire extinguishing. When the wind speed is high, the participation of helicopters in forest fires should be reduced.

Helicopters need to be equipped with fire-fighting materials such as water and dry powder to effectively extinguish fires. There is a limit to how much fire-fighting material a helicopter can carry. And forest fires generally wide range, far away from the ignition point, is not conducive to the deployment of helicopters to fight forest fires. In order to better fight forest fires, helicopters and fire-fighting materials need to be evenly distributed in multiple areas of the forest. At the same time, the impact of forest fires on forest ecology and infrastructure needs to be considered when deploying helicopters [32]. You can't put all the fire fighting helicopters in one area. So the helicopter can get as close to the fire as possible.

If a forest fire occurs, technical and personal involvement in fighting the fire is essential. For this reason, we must not forget to develop new and more efficient firefighting equipment and materials, such as fire hose systems, rescue tools, flame inhibitors and so on. As well as fighting forest fires, prevention, detection and early warning of forest fires are also an important part of forest fire control. In order to reduce the loss of forest fire, the risk of fire occurrence needs to be reduced at the source. When the forest terrain is complex, it is difficult for personnel to enter the forest for monitoring work, and there are fewer residents near the forest. This is mainly because mountainous terrain often has complex terrain and rugged ground, which makes the deployment of firefighting personnel and equipment difficult. For example, ridges and steep slopes can prevent firefighters from reaching areas where fires are spreading fastest. In addition, the mountainous terrain can also cause the fire to spread faster, because the fire can travel down the slope quickly. Canyon terrain is often narrow and deep, which can cause a fire to be compressed into a relatively enclosed space, making the fire more intense and difficult to control. In addition, the wind direction and speed of the canyon's terrain can cause the fire to spread in a particular direction, making it more difficult to fight the fire. Drobyshev et al. [33] confirmed this in a study of the Sedney National Wildlife Refuge in eastern

Michigan. At this point, the benefits of deploying remote monitoring personnel are smaller. Helicopters should be used in such cases for greater benefit.

## 5. Conclusion

This article assumes that forest fires can be controlled by remote monitoring, personnel firefighting and helicopter firefighting. Considering that forest fire is very easy to spread, this article constructs three differential game models. And the equilibrium results are compared and analyzed. The study of this article shows that the stronger the wind, the more residents need to flee. However, strong winds are not conducive to crews and helicopters fighting the fire. Rather than fighting fires, residents are more inclined to detect forest fires in time through remote monitoring. When the personnel can effectively control the fire, the personnel fire extinguishing mode is preferentially selected. Otherwise, helicopter firefighting mode should be selected.

The study of this article can also be extended to some extent. For example, this article only considers the situation that there are more residents near the forest, the decision-making of the government and residents is constantly changing, and the country has a high level of economic development. In future studies, relevant issues can be studied when residents are far away from the forest, residents make decisions according to government instructions, and the state must accept assistance. In addition, the research of this article is not only applicable to the problem of forest fire suppression, but also has certain reference significance for forest pests, forest floods and other related problems.

## Supporting information

**S1 File.**
(DOCX)

## Author Contributions

**Conceptualization:** Yuntao Bai.

**Data curation:** Yuntao Bai.

**Formal analysis:** Yuntao Bai.

**Funding acquisition:** Lan Wang.

**Investigation:** Lan Wang.

**Methodology:** Lan Wang.

**Software:** Xiaolong Yuan.

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
