## [Decision Letter · Decision Letter 0]

22 Jun 2023

PONE-D-23-13225Remote Monitoring, Personnel Extinguishment or Helicopter Extinguishment? How to Control Forest Fires More EffectivelyPLOS ONE

Dear Dr. Wang,

Thank you for submitting your manuscript to PLOS ONE. After careful consideration, we feel that it has merit but does not fully meet PLOS ONE’s publication criteria as it currently stands. Therefore, we invite you to submit a revised version of the manuscript that addresses the points raised during the review process.

We look forward to receiving your revised manuscript.

Kind regards,

Sathishkumar V E

Academic Editor

PLOS ONE

Journal Requirements:

2. PLOS requires an ORCID iD for the corresponding author in Editorial Manager on papers submitted after December 6th, 2016. Please ensure that you have an ORCID iD and that it is validated in Editorial Manager. To do this, go to ‘Update my Information’ (in the upper left-hand corner of the main menu), and click on the Fetch/Validate link next to the ORCID field. This will take you to the ORCID site and allow you to create a new iD or authenticate a pre-existing iD in Editorial Manager. Please see the following video for instructions on linking an ORCID iD to your Editorial Manager account: https://www.youtube.com/watch?v=_xcclfuvtxQ.

Reviewers' comments:

Reviewer's Responses to Questions

**Comments to the Author**

1. Is the manuscript technically sound, and do the data support the conclusions?

Reviewer #1: Yes

Reviewer #2: Yes

2. Has the statistical analysis been performed appropriately and rigorously? 

Reviewer #1: Yes

Reviewer #2: Yes

3. Have the authors made all data underlying the findings in their manuscript fully available?

Reviewer #1: Yes

Reviewer #2: Yes

4. Is the manuscript presented in an intelligible fashion and written in standard English?

Reviewer #1: Yes

Reviewer #2: Yes

5. Review Comments to the Author

Reviewer #1: The paper describes an interesting solution how to control forest fires more effectively.

We agree with the authors and the results that the focus of fire management has moved from direct firefighting to fire prevention in recent years, as evidenced by the results of the submitted paper. For the very fires that have still occur despite prevention, it is necessary to have a suitable technique. Prevention is highly important but current climatic conditions and their demonstrable changes give rise to extreme fires. Forest fire protection is highly topical issue globally. An overall goal within prevention is to minimize the risk of disaster (which definitely includes forest fires) occurrence, as the environmental loss is incalculable.

In the description of problem (3), you mention the deployment of helicopters as the most effective way of extinguishing forest fires. In the case of forests, the availability, terrain and water source must be taken into account. So it is not always the most effective way of extinguishing forest fires with the help of helicopters. I suppose that you meant the described problem (3) for the conditions mentioned above (complex terrain, slope, insufficient structure of forest roads).

For a better orientation in the results, I would like to ask for a more detailed explanation of the parameter income generated by unit personnel input.

The results of the presented article confirm the established trend of remote monitoring as a basis for forest fire prevention and protection. The differential game model of forest fire management showed possible scenarios of management and protection of the population and the environment against possible adverse effects of fires in the forest environment.

In the results, you mention the sentence "The role of prevention, detection and early warning is always greater than the significance of extinguishing the fire after it occurs". I can partially agree with this statement, but on the other hand, if a fire starts, technique and personal participation in extinguishing it is essential. For this reason, we cannot forget the development of new, more efficient firefighting equipment. The role of prevention, detection and early warning should therefore not be greater, but should be an important part of the entire fire protection system.

Reviewer #2: 1. The abstract provides a clear and concise summary of the study's objectives and methodologies. It effectively highlights the importance of controlling forest fires and presents the three approaches to be analyzed.

2. It would be beneficial to provide more context regarding the significance of the research. Why is it important to investigate different methods of controlling forest fires? How can the findings contribute to existing knowledge in the field? Adding a brief statement on the practical implications or potential benefits of the study would strengthen the abstract.

3. The Manuscript briefly mentions the impact of wind speed and terrain complexity on the difficulty of extinguishing forest fires. It would be helpful to expand on this point and explain how these factors are taken into account in the differential game model.

4. The abstract mentions a conclusion but does not provide any specific details about the findings or the comparative analysis conducted. It would be valuable to include a sentence or two summarizing the main conclusions of the study.

5. The use of high-tech equipment in remote monitoring of forest fires is mentioned in the abstract. It would be useful to elaborate on the potential advantages or specific technologies that could be employed in this context.

6. The Manuscript could benefit from a clearer statement on the social benefit aspect mentioned in the keywords. What are the social benefits associated with different modes of fire control, and how are they assessed in the study?

7. It would be helpful to include the scope or geographical focus of the research. Is the study specifically focused on a certain region or is it more general in nature? Clarifying this aspect would provide readers with a better understanding of the study's applicability.

8. Overall, the Manuscript effectively outlines the study's objectives and methodologies. However, further expansion on the research's significance, specific findings, and practical implications would enhance its clarity and appeal to potential readers.

9. The authors are advised, in order to validate the data set they have identified, to broaden the conclusions with a brief comparison between their values and those of the other studies (cited by themselves in the references) who faced the same time-range prediction on the same region.

10. Rewrite Conclusion again with main findings

6. PLOS authors have the option to publish the peer review history of their article (what does this mean?). If published, this will include your full peer review and any attached files.

Reviewer #1: **Yes: **Richard Hnilica

Reviewer #2: **Yes: **Suraj Kumar Singh

---

## [Author Response · Author response to Decision Letter 0]

4 Jul 2023

Response to reviewer1

Dear Editors and Reviewers:

Many thanks for your valuable comments and suggestions on our manuscript entitled “Remote Monitoring, Personnel Extinguishment or Helicopter Extinguishment? How to Control Forest Fires More Effectively” (Manuscript ID: PONE-D-23-13225). The comments and suggestions are very helpful for improving our paper. We have made revision based on the comments and suggestions. Please find our response as follows, and we have made revision which marked in blue in the paper. Attached please find the revised version, which we would like to submit for your kind consideration.

Point 1：

In the description of problem (3), you mention the deployment of helicopters as the most effective way of extinguishing forest fires. In the case of forests, the availability, terrain and water source must be taken into account. So it is not always the most effective way of extinguishing forest fires with the help of helicopters. I suppose that you meant the described problem (3) for the conditions mentioned above (complex terrain, slope, insufficient structure of forest roads).

Response 1: 

Thank you very much for your suggestion. In the revised version, the constraints of helicopter availability, terrain and water source are described, and the helicopter is a very effective fire extinguishing method under certain conditions. For details, see lines 157 to 169 in blue.

Point 2：

For a better orientation in the results, I would like to ask for a more detailed explanation of the parameter income generated by unit personnel input.

Response 2: 

Thank you very much for your suggestion. In the revised version, the parameter income generated by unit personnel input is elaborated in detail in this paper, so that readers have a clearer understanding of it. For details, see lines 215 to 231 in blue.

Point 3：

 In the results, you mention the sentence "The role of prevention, detection and early warning is always greater than the significance of extinguishing the fire after it occurs". I can partially agree with this statement, but on the other hand, if a fire starts, technique and personal participation in extinguishing it is essential. For this reason, we cannot forget the development of new, more efficient firefighting equipment. The role of prevention, detection and early warning should therefore not be greater, but should be an important part of the entire fire protection system.

Response 3: 

Thank you very much for your suggestion. In the revised version, this article has changed the relevant content . This article illustrates “As well as fighting forest fires, prevention, detection and early warning of forest fires are also an important part of forest fire control.”. At the same time, this article also expounds “more efficient firefighting equipment and materials”. For details, see lines 518 to 533 in blue.

Response to reviewer2

Dear Editors and Reviewers:

Many thanks for your valuable comments and suggestions on our manuscript entitled “Remote Monitoring, Personnel Extinguishment or Helicopter Extinguishment? How to Control Forest Fires More Effectively” (Manuscript ID: PONE-D-23-13225). The comments and suggestions are very helpful for improving our paper. We have made revision based on the comments and suggestions. Please find our response as follows, and we have made revision which marked in blue in the paper. Attached please find the revised version, which we would like to submit for your kind consideration.

Point 1：

The abstract provides a clear and concise summary of the study's objectives and methodologies. It effectively highlights the importance of controlling forest fires and presents the three approaches to be analyzed.

Response 1: 

Yes, Thank you for your comments on the abstract. The abstract provides a clear and concise summary of the study's objectives and methodologies. It effectively highlights the importance of controlling forest fires and presents the three approaches to be analyzed.

Point 2：

It would be beneficial to provide more context regarding the significance of the research. Why is it important to investigate different methods of controlling forest fires? How can the findings contribute to existing knowledge in the field? Adding a brief statement on the practical implications or potential benefits of the study would strengthen the abstract.

Response 2: 

Thank you very much for your suggestion. In the revised version, this paper briefly explains why different forest fire fighting methods are very important in the abstract. For details, see lines 13 to 14 in blue. In the revised version of the manuscript, the reasons for the importance of different methods of forest fire control are also described in detail. For details, see lines 61 to 73 in blue.

In this abstract, we briefly describe how our findings contribute to existing knowledge in this field. For details, see lines 16 to 18 in blue. In a revised version of the manuscript, the paper also provides a detailed description of how these findings contribute to existing knowledge in the field. For details, see lines 104 to 109 in blue.

Point 3：

The Manuscript briefly mentions the impact of wind speed and terrain complexity on the difficulty of extinguishing forest fires. It would be helpful to expand on this point and explain how these factors are taken into account in the differential game model.

Response 3: 

Thank you very much for your suggestion. In the revised version of the manuscript, in "2.1.3 Variable definition", this paper elaborated the influence of wind speed and terrain complexity on the difficulty of extinguishing forest fires. For details, see lines 232 to 241 in blue. At the same time, "2.2" details how to take these factors into account in differential game models. For details, see lines 295 to 302 in blue. 

Point 4：

The abstract mentions a conclusion but does not provide any specific details about the findings or the comparative analysis conducted. It would be valuable to include a sentence or two summarizing the main conclusions of the study.

Response 4: 

Thank you very much for your suggestion. In the revised version, this paper rewrites the research conclusions. The re-conclusion may reflect the specific details of the research findings or carry out a comparative analysis. For details, see lines 20 to 24 and 540 to 544 in blue. 

Point 5：

The use of high-tech equipment in remote monitoring of forest fires is mentioned in the abstract. It would be useful to elaborate on the potential advantages or specific technologies that could be employed in this context.

Response 5: 

Thank you very much for your suggestion. In the revised version, this paper introduces in detail the application of high-tech equipment in forest fire remote monitoring. And potential advantages or specific technologies that may be employed in this regard. For details, see lines 134 to 145 in blue. 

Point 6：

The Manuscript could benefit from a clearer statement on the social benefit aspect mentioned in the keywords. What are the social benefits associated with different modes of fire control, and how are they assessed in the study?

Response 6: 

Thank you very much for your suggestion. In the revised version, the article makes a more explicit statement on the social benefits mentioned in the keywords. For details, see lines 242 to 254 in blue. In the revised version, the paper also makes it clearer how to evaluate the social benefits. For details, see lines 255 to 265 in blue. 

Point 7：

It would be helpful to include the scope or geographical focus of the research. Is the study specifically focused on a certain region or is it more general in nature? Clarifying this aspect would provide readers with a better understanding of the study's applicability.

Response 7: 

Thank you very much for your suggestion. In the revised version, the scope of the study is more clearly stated. The scope of application of this study is all areas of the earth with forest. For details, see lines 121 to 128 in blue. 

Point 8：

Overall, the Manuscript effectively outlines the study's objectives and methodologies. However, further expansion on the research's significance, specific findings, and practical implications would enhance its clarity and appeal to potential readers.

Response 8: 

Thank you very much for your suggestion. In the revised version of the manuscript, this paper further expands the significance, specific findings and practical significance of the study. For details, see lines 61 to 73 in blue and lines 104 to 109 in blue.

Point 9：

The authors are advised, in order to validate the data set they have identified, to broaden the conclusions with a brief comparison between their values and those of the other studies (cited by themselves in the references) who faced the same time-range prediction on the same region.

Response 9: 

Thank you very much for your suggestion. In the modified version, for the data set identified, their values were compared to other studies facing the same time frame predictions in the same region. The corresponding references are also cited. For details, see lines 484 to 486, 492 to 496, 502 to 507, 524 to 533 in blue.

Point 10：

Rewrite Conclusion again with main findings

Response 10: 

Thank you very much for your suggestion. In the revised version, this paper rewrites the research conclusions. For details, see lines 20 to 24 and 540 to 544 in blue.

---

## [Decision Letter · Decision Letter 1]

26 Jul 2023

Remote Monitoring, Personnel Extinguishment or Helicopter Extinguishment? How to Control Forest Fires More Effectively

PONE-D-23-13225R1

Dear Dr. Wang,

We’re pleased to inform you that your manuscript has been judged scientifically suitable for publication and will be formally accepted for publication once it meets all outstanding technical requirements.

Kind regards,

Sathishkumar V E

Academic Editor

PLOS ONE

Additional Editor Comments (optional):

Reviewers' comments:

Reviewer's Responses to Questions

**Comments to the Author**

1. If the authors have adequately addressed your comments raised in a previous round of review and you feel that this manuscript is now acceptable for publication, you may indicate that here to bypass the “Comments to the Author” section, enter your conflict of interest statement in the “Confidential to Editor” section, and submit your "Accept" recommendation.

Reviewer #1: All comments have been addressed

Reviewer #2: All comments have been addressed

2. Is the manuscript technically sound, and do the data support the conclusions?

Reviewer #1: Yes

Reviewer #2: Yes

3. Has the statistical analysis been performed appropriately and rigorously? 

Reviewer #1: Yes

Reviewer #2: Yes

4. Have the authors made all data underlying the findings in their manuscript fully available?

Reviewer #1: Yes

Reviewer #2: Yes

5. Is the manuscript presented in an intelligible fashion and written in standard English?

Reviewer #1: Yes

Reviewer #2: Yes

6. Review Comments to the Author

Reviewer #1: The authors of the article responded sufficiently to my comments and incorporated them into the text. After the mentioned adjustments, I recommend publishing the submitted article in the journal PLOS ONE.

Reviewer #2: In this article, the author addresses the critical issue of forest fire control and explores three main methods: remote monitoring, personnel extinguishing, and helicopter extinguishment. The primary focus of the article lies in optimizing resource allocation and guiding policy formulation, rather than just the technical aspects of fire elimination.

The use of a differential game model to analyze the equilibrium result and conduct a comparative analysis is a novel approach that sets this article apart from others. This theoretical framework allows for a deeper understanding of how these fire control methods interact and influence one another under various conditions.

The author's acknowledgment of the ecological impact of forest fires and the importance of protecting the forests' ecological functions is commendable. As forest fires are heavily influenced by weather conditions and terrain complexity, it is essential to have effective management methods in place to mitigate the risks and protect the environment for long-term development and economic efficiency.

The conclusion drawn by the author, highlighting the significance of early detection through remote monitoring and the preference for personnel fire extinguishing when effectively controllable, is logical and supported by the theoretical analysis. The consideration of wind speed as a factor affecting the residents' need to evacuate and the firefighting efforts by crews and helicopters adds depth to the overall findings.

7. PLOS authors have the option to publish the peer review history of their article (what does this mean?). If published, this will include your full peer review and any attached files.

Reviewer #1: **Yes: **Richard Hnilica

Reviewer #2: No

---

## [Editor Report · Acceptance letter]

31 Jul 2023

PONE-D-23-13225R1 

Remote Monitoring, Personnel Extinguishment or Helicopter Extinguishment? How to Control Forest Fires More Effectively 

Dear Dr. Wang:

I'm pleased to inform you that your manuscript has been deemed suitable for publication in PLOS ONE. Congratulations! Your manuscript is now with our production department. 

Kind regards, 

on behalf of

Dr. Sathishkumar V E 

Academic Editor

PLOS ONE